# Predicting the Potential Distribution of *Hypericum perforatum* under Climate Change Scenarios Using a Maximum Entropy Model

**DOI:** 10.3390/biology13060452

**Published:** 2024-06-19

**Authors:** Yulan Hao, Pengbin Dong, Liyang Wang, Xiao Ke, Xiaofeng Hao, Gang He, Yuan Chen, Fengxia Guo

**Affiliations:** 1College of Agronomy, Gansu Provincial Key Laboratory of Good Agricultural Production for Traditional Chinese Medicines, Gansu Provincial Engineering Research Centre for Medical Plant Cultivation and Breeding, Gansu Agricultural University, Lanzhou 730070, China; haoyl@st.gsau.edu.cn (Y.H.); dongpb@stumail.nuw.edu.cn (P.D.); w18856278265@163.com (L.W.); 2Sichuan Kanghong Chinese Herbal Medicine Planting Co., Ltd., Chengdu 611930, China; txzz@cnkh.com (X.K.); cygcx1963@163.com (X.H.); 021757@cnkh.com (G.H.); 3College of Life Science and Technology, Gansu Provincial Key Laboratory of Arid Land Crop Science, Gansu Key Laboratory of Crop Genetic and Germplasm Enhancement, Gansu Agricultural University, Lanzhou 730070, China

**Keywords:** *Hypericum perforatum*, MaxEnt model, suitable distribution area, climate change, GIS technology

## Abstract

**Simple Summary:**

As the modern medical industry advances, *Hypericum perforatum*, with its excellent medicinal properties, has become well-known to international researchers. However, due to the large-scale exploitation and use of its wild resources, the population of *H. perforatum* has dramatically decreased. Artificial cultivation often faces issues such as unstable yields and varying concentrations of effective components, making it difficult for the current *H. perforatum* resources to satisfy market demands, necessitating an expansion in its cultivation area. Therefore, this study uses *H. perforatum* as the research material and predicts its suitable distribution range from the perspective of ecological niche theory. By identifying the dominant environmental factors that influence the growth of *H. perforatum*, this research provides a scientific basis for the protection and sustainable use of its wild resources.

**Abstract:**

*H. perforatum*, as one of the Traditional Chinese Medicinal materials, possesses a variety of pharmacological activities and high medicinal value. However, in recent years, the wild resources of *H. perforatum* have been severely depleted due to global climate change and human activities, and artificial cultivation faces problems such as unstable yield and active ingredient content. This poses a serious obstacle to the development and utilization of its resources. Therefore, this experiment took *H. perforatum* as the research object and used 894 distribution records of *H. perforatum* and 36 climatic environmental factors, using the MaxEnt model and GIS technology to explore the main climatic factors affecting the distribution of *H. perforatum*. Additionally, by utilizing the principles of ecological niche theory, the potential suitable distribution regions of *H. perforatum* across past, present, and future timelines were predicted, which can ascertain the dynamics of its spatial distribution patterns and the trend of centroid migration. The results indicate that the main environmental factors affecting the geographical distribution of *H. perforatum* are solar radiation in April (Srad4), solar radiation in September (Srad9), mean temperature of driest quarter (Bio9), solar radiation in November (Srad11), annual mean temperature (Bio1), and annual precipitation (Bio12). Under future climate scenarios, there is a remarkable trend of expansion in the suitable distribution areas of *H. perforatum*. The centroid migration indicates a trend of migration towards the northwest direction and high-altitude areas. These results can provide a scientific basis for formulating conservation and sustainable use management strategies for *H. perforatum* resources.

## 1. Introduction

Medicinal plants, as one of the important sources of Traditional Chinese Medicine play a significant role in the fight against diseases. The growth, development, reproduction, and ecological distribution of each medicinal plant are closely related to ecological factors such as climate, soil, and topography. Suitable ecological factors are conducive to the growth and development of medicinal plants, as well as the production and accumulation of active ingredients, thereby improving the quality of medicinal materials [1].

Climate itself is a complex natural phenomenon, which is influenced by the combined effects of various factors during its formation and evolution, including the interactions between human activities and natural factors. These interactions lead to varying degrees of global climatic changes [2]. Since the beginning of the 20th century, the global average temperature has risen by nearly 1 °C [3]. The IPCC’s Sixth Assessment Report indicates that by the year 2100, the global average temperature is projected to increase by a minimum of 0.3–1.7 °C and a maximum of 2.6–4.8 °C [4,5]. Climate change, characterized by rising temperatures, has become an indisputable objective fact [6]. Extensive research indicates that a significant driving factor behind the sharp decline in biodiversity is climate change, as it may lead to a reduction in the suitable distribution areas for species [7,8], fragmentation and loss of original distribution areas, decreased richness, reduced suitable habitats, species reproductive isolation, and increased frequency of pest outbreaks. At the macro scale, temperature, precipitation, light, and soil are the most important environmental factors influencing species distribution and biodiversity [9]. Among these factors, temperature serves as the heat source for seed germination and the subsequent growth and development of medicinal plants. Precipitation is essential for maintaining normal photosynthesis and other metabolic processes in plants. Insufficient changes in light intensity and photoperiod can lead to poor plant development. Soil is also an indispensable condition for the growth of medicinal plants. The chemical properties of soil are one of the main factors affecting the availability of soil nutrients and the chemical substances within the soil [10]. Many species will seek new regions with climate conditions most suitable for their growth in order to adapt to the development of climate change, and changes in their distribution patterns directly reflect climate change [11]. In conclusion, effectively predicting the impact of climate change on the potential geographic distribution of species has become an important topic in current geographical, ecological, and global change research [12].

At present, research on predicting the impact of climate change on species distribution is gaining increasing attention from researchers. Species Distribution Models (SDMs), also known as ecological niche models, originated from the study of the relationship between plant communities and environmental gradients. The main principle is to associate known species distribution data with environmental data using specific algorithms, to obtain the relationship between species distribution and environmental variables. This relationship is then used to simulate and predict the potential distribution range of target species in a specified area and to map the distribution of species in geographical space and certain conditions in the past, present, or future [13]. With the rapid development of Geographic Information Systems (GISs) and increasing ease of access to various data such as climate, topography, and human impact indices, the application capabilities of Species Distribution Models (SDMs) have been greatly enhanced. Currently, SDMs have become important tools in basic ecology and biogeography research and are widely used to address various issues in ecology, biogeography, biological evolution, species invasion [14], conservation biology, and climate change research. They are extensively utilized for locating rare or endangered species, planning new protected areas [15], assessing the impact of human activities on biodiversity, predicting the effects of climate change on species distribution [16], preventing the spread of invasive species, identifying disease vectors, understanding the non-biological needs of species, and improving agricultural productivity, among other areas.

Multiple Species Distribution Models are used to predict species distribution based on the relationship between species and environmental variables. Among these, the MaxEnt model is an ideal model that combines species distribution records with environmental factors to predict species distribution. Compared to other models, the MaxEnt model requires a low number of samples and has high accuracy and stability, even with few sample points [17]. Therefore, this study takes *H. perforatum* as the research object, based on its distribution records and related environmental factors, and uses a combination of the MaxEnt model and GIS technology to study the ecological suitability of *H. perforatum* in China, to predict the distribution trends and characteristics of *H. perforatum* in different periods, and to explore the leading climatic factors affecting its distribution. This provides an important scientific basis for the resource conservation, utilization, and site selection for artificial cultivation of *H. perforatum*.

*H. perforatum*, also known as “guan ye lian qiao” [18], is a perennial herbaceous medicinal plant in the Hypericum genus of the Garcinia family, locally referred to as “guo lu huang”, “xiao zhong huang”, “qian ceng lou”, and known as St. John’s wort in Western countries. Hypericum is the largest genus in the family Hypericaceae, with approximately 460 species worldwide, divided into 30 sections and distributed globally. In China, there are 55 species and 8 subspecies, with two species being of uncertain status [19]. The Hypericum species in China are classified into eight sections: Sect. *Ascyreia* Choisy, Sect. Takasagoy (Y. Kimura) N. Robson, Sect. *Roscyna* (Spach) R. Keller, Sect. *Spachium* (R. Keller) N. Robson, Sect. *Taeniocarpium* Jaub. & Spach, Sect. *Hirtella* Stapf, Sect. *Adenosepalum* Spach, and Sect. *Hypericum*. The most representative active substances in Hypericum are naphthodianthrone derivatives, mainly including hypericin and pseudohypericin, which possess antidepressant [20], antiviral, and immune-enhancing properties [21]. These compounds also exhibit significant anti-DNA and RNA virus activities, making them useful in the treatment of AIDS, with more pronounced effects and no side effects compared to other similar structures [22].

*H. perforatum* grows in mountain shrublands and grasslands, as well as roadside areas, at altitudes between 800 and 2100 m. It prefers warm, moist environments and is cold-resistant, belonging to the temperate flora. Its distribution spans across Asia, Europe, Africa, and South America, with its primary presence in the provinces of Sichuan, Shaanxi, Gansu, Shandong, Hebei, Shanxi, Jiangsu, Jiangxi, Henan, Hubei, Hunan, and Guizhou in China [23]. It is considered cool in nature, with a pungent flavor, and is associated with the liver meridian. It has been used to dispel liver depression, clear heat and remove dampness, reduce swelling, and promote lactation [24]. Modern pharmacological research has shown that extracts of *H. perforatum* have pharmacological properties such as antidepressant [25], antibacterial [26], antiviral [27], antioxidant [28], and anticancer [29] effects. With the expanding scale of the development and utilization of *H. perforatum*, its excellent medicinal value has become increasingly prominent, earning its place in the Traditional Chinese Medicine market. Since being included in the Pharmacopoeia of the People’s Republic of China in 2005, it has garnered increasing attention from the academic community and has subsequently been included in the pharmacopoeias of European and American countries such as Poland, Switzerland, Romania, and the United States. It has become one of the best-selling plants in the world, gradually emerging as another hot research topic in medicinal plants following *Panax ginseng*, *Rubus idaeus*, *Artemisia annua*, *Taxus wallichiana* var. *chinensis*, and *Catharanthus roseus* [30]. Because the main active component (Hypericin) is primarily found in flowers and leaves, it is customary to harvest during the blooming period, which interrupts the flowering and fruiting process, further impacting the plant’s reproduction. Additionally, extensive exploitation and utilization of wild resources have led to a drastic reduction in the population of *H. perforatum*, resulting in a gradual depletion of wild resources. Artificial cultivation often faces instability in yield and active ingredient concentration due to low propagation rates, long growth cycles, and vulnerability to pests and diseases [31]. Therefore, meeting market demands with the existing *H. perforatum* resources has become challenging, and expanding the cultivation area of this plant is imperative. Therefore, this study predicts the suitable distribution range of *H. perforatum* from the perspective of ecological niche theory, selects the dominant environmental factors affecting its growth, understands the spatial distribution pattern changes and centroid migration trends, and aims to avoid the waste of human, financial, and material resources caused by introduction without theoretical support. This study provides a theoretical basis for the sustainable development of *H. perforatum* resources.

## 2. Materials and Methods

### 2.1. Species Distribution Data

The geographic distribution data of *H. perforatum* used in this study were sourced from field surveys, the Chinese Virtual Herbarium (CVH, https://www.cvh.ac.cn/ (accessed on 14 October 2023)), and related literature records. To ensure data accuracy, distribution records without specific geographic information were deleted, and for the sampling points with detailed geographic locations but no latitude and longitude information, Google Maps was used to locate and obtain accurate latitude and longitude information. Spatially overlapping data points within the 5 km range were discarded through ENMTools_1.4.4 software, thus avoiding overfitting in the model caused by repeatedly distributed points. Finally, a total of 894 valid data on distribution information were obtained (Appendix A), including 24 records from field collection, 627 records from Chinese Virtual Herbarium, and 243 records from the literature. ArcGIS10.8 (https://www.32r.com/soft/94654.html (accessed on 23 November 2023)) was used to draw the geographic distribution map of *H. perforatum*. (Figure 1).

### 2.2. Bioclimatic Data Acquisition and Screen

A total of 36 environmental factors that may affect *H. perforatum* were selected (Appendix A), which included 19 climatic factors with a spatial resolution of 2.5 arc-min and 12 radiation factors downloaded from the WorldClim database (http://www.worldclim.org (accessed on 23 October 2023)). Elevation data were obtained from the Geospatial Data Cloud (http://www.gscloud.cn (accessed on 23 October 2023)). Additionally, 4 soil data points—soil pH (t_pH), soil electrical conductivity (t_ece), soil organic carbon content (t_oc), and soil available water capacity (Awc)—were sourced from the Harmonized World Soil Database (http://www.fao.org/soils-portal/soil-survey (accessed on 24 October 2023)). The data are the Last inter-glacial (LIG, (120,000−140,000 years)), Last glacial maximum (LGM, (22,000 years)), Mid holocene (MH, (6000 years)), current (1970−2000) period, and future periods (the 2050s, 2090s). The climate data for the LIG, the LGM, and the MH were used for simulating geographic distribution patterns, which were developed by the Climate Community System Model version 4.0 (CCSM4.0) created by the National Center for Atmospheric Research. The corresponding environmental data for the periods of the 2050s (2041–2060) and the 2090s (2081–2100) were obtained from four sets of projections, SSP1-2.6, SSP2-4.5, SSP3-7.0, and SSP5-8.5, under the BCC-CSM2-MR climate model released by the IPCC Sixth Assessment Report.

To avoid the impact of multicollinearity among influencing factors on the simulation results and to ensure the accuracy of the model’s predicted outcomes, the initial step involved importing 36 environmental factors and 894 distribution data points into the MaxEnt 3.4.4 ecological niche modeling software for computation. After eliminating environmental factors with a contribution rate of zero, the calculation was repeated until all remaining environmental factors had a non-zero contribution rate. Subsequently, using the sampling function of ArcGIS 10.8 software to extract environmental data from each distribution point, SPSS 27.0 was employed for correlation analysis. When the correlation coefficient values between two environmental factors exceeded 0.8, the environmental factor with the smaller contribution rate was discarded (Appendix A). For environmental variables, eight climate variables (Srad4, Srad9, Bio9, Srad11, Srad10, Bio1, Bio3, and Bio12) were selected to build the model (Table 1).

### 2.3. Settings of the MaxEnt Model

Use the MaxEnt model to construct a relationship model between the geographic distribution of *H. perforatum* and environmental factors. Set 25% of the distribution points as the test data and 75% as the training data; the number of repetitions is set to 100, with the Receiver Operating Characteristic (ROC) curve, Jackknife test, and response curves selected; the software’s output file format is logistic, with the output file type as ASC, and all other parameters set to the software’s default values. Evaluate the accuracy of the model’s predictive results with the area under the ROC curve (AUC) [32]; use the Jackknife test to detect the importance of environmental variables, and assess the suitability range of environmental factors through the response curves.

### 2.4. Model Precision Test and Suitability Level Classification

The accuracy of the MaxEnt model was verified using the Receiver Operator Characteristic (ROC), which was expressed as the area under the curve (AUC). AUC is a comprehensive evaluation criterion that reflects both the accuracy and specificity of the ROC curve. The range of AUC values is 0–1, with values closer to 1 indicating higher precision of the model’s predictive results. AUC values in the range of 0.1–0.6 represent a failure in the model’s predictive results, 0.6–0.7 indicate poor predictive effects, 0.7–0.8 indicate average predictive results, 0.8–0.9 represent good predictive results, and 0.9–1.0 indicate excellent predictive effects [33].

The MaxEnt model’s prediction result was imported into ArcGIS 10.8, and the SDM tool was used to convert it into Raster data for reclassification and habitat suitability zoning. The computation results classified the suitable habitat of *H. perforatum* into four levels: unsuitable habitat (0–0.1), low suitability habitat (0.1–0.3), medium suitability habitat (0.3–0.5), and high suitability habitat (>0.5).

### 2.5. Analysis of Centroid Migration in Suitable Distribution Areas

Suitable habitats for *H. perforatum* were defined as spatial units with a species presence probability greater than 0.50, while habitats unsuitable for *H. perforatum* were defined as units with a probability less than 0.50, leading to the creation of a presence–absence (0, 1) matrix under different future climate scenarios. Based on the (0, 1) matrix, using the Raster Calculator, following the step of “Toolbox-Spatial Analyst Tools-Map Algebra”, further analyze the pattern changes in suitable habitats for *H. perforatum* under various future climate scenarios. The future area changes were calculated based on the current area of *H. perforatum* suitable habitats. Lastly, the matrix change values were loaded into the ArcGIS software to examine the spatial pattern changes in *H. perforatum* suitable habitats in the future.

To consider the suitable habitats for *H. perforatum* on a holistic level, the study simplified it to a vector particle (centroid) and used changes in the centroid position to reflect the migration direction of suitable habitats for the species. The centroid of the *H. perforatum* suitable habitat distribution was defined as the geometric center of the grid cells with a species distribution probability value greater than 0.50. The “SDMtools” in ArcGIS software were used to analyze the centroids of the past, current, and future suitable habitats and to compare the trends in centroids and suitable habitat changes across different suitable areas. Finally, Bigemap 15.4 software was employed to calculate the migration distance of the centroids for *H. perforatum* under different climate conditions.

## 3. Results

### 3.1. Accuracy of Model Analysis

Through 100 iterations of the MaxEnt model, combining the geographic distribution data of *H. perforatum* with environmental factors, the results show an average AUC value of 0.936 (Figure 2), which is greater than 0.90. This indicates that the model’s predictive results are highly reliable and can accurately predict the potential suitable distribution areas of *H. perforatum* under climate change.

### 3.2. Dominant Environmental Factors

Table 1 shows among the solar radiation of the 12 months, the April solar radiation (Srad4) and September solar radiation (Srad9) have the greatest impact on the distribution of *H. perforatum*, with a combined contribution rate reaching 44.10% and a combined sum of permutation importance of 15.50%. Compared to April and September solar radiation, the November solar radiation (Srad11) and October solar radiation (Srad10) have a smaller impact on the distribution of *H. perforatum*, with contribution rates of 13.20% and 9.70%, respectively, and permutation importance of 10.40% and 5.60%, respectively. Among the 19 climate factors, the mean temperature of the driest quarter (Bio9), the annual mean temperature (Bio1), isothermality (Bio3), and annual precipitation (Bio12) have a larger influence on the suitable distribution of *H. perforatum*, with a combined contribution rate of 33.10% and a combined sum of permutation importance of 68.40%. According to the jackknife test results, as shown in Figure 3, the weights of April solar radiation (Srad4), September solar radiation (Srad9), the mean temperature of the driest quarter (Bio9), and annual precipitation (Bio12) are relatively high.

The single-factor response curve characterizes the relationships between the presence probability of the predicted species and each environmental variable, which can clearly reveal the correlation and trend between the existence probability of the suitable areas for *H. perforatum* and the dominant environmental factors. When the existence probability is greater than 0.5, it indicates that the ecological factor values are suitable for plant growth [34]. Based on the results of the Jackknife test and the contribution rates of environmental variables, it was found that the April solar radiation (Srad4), September solar radiation (Srad9), mean temperature of the driest quarter (Bio9), November solar radiation (Srad11), annual mean temperature (Bio1), and annual precipitation (Bio12) have the greatest impact on the distribution of *H. perforatum*. Single-factor modeling was carried out based on these six dominant environmental factors, and the single-factor response curves were drawn, as shown in Figure 4. The results indicate that the optimal solar radiation range for April (Srad4) is 14,578.68–15,516.58 kJ m^−2^ day^−1^ (Figure 4a); the optimal solar radiation range for September (Srad9) is 9672.00–11,340.02 kJ m^−2^ day^−1^ (Figure 4b); the optimal range for the mean temperature of the driest quarter (Bio9) is 0.59–2.62 °C (Figure 4c); the optimal solar radiation range for November (Srad11) is 7977.24–9003.09 kJ m^−2^ day^−1^ (Figure 4d); the optimal temperature range for the annual mean temperature (Bio1) is 9.80–16.16 °C (Figure 4e); and the optimal annual precipitation range (Bio12) is 632.77–1288.38 mm (Figure 4f).

### 3.3. Ecological Niche Modeling

#### 3.3.1. Suitable Areas under Past Climate Conditions

Figure 5 and Table 2 show under past climate scenarios, the high and medium suitability areas for *H. perforatum* changed significantly. The low-suitability areas were mainly distributed in regions such as Shaanxi Province, Shanxi Province, Henan Province, Hubei Province, and Anhui Province. Table 3 shows that compared to the present, the total suitable area for *H. perforatum* under past climate conditions was significantly larger than that under contemporary climate conditions. The total suitable area and the area of high suitability have shown a decreasing trend from the LIG to the present. The total suitable area and the high suitability area reached their maximum during the LIG at 309.18 × 10^4^ km^2^ and 104.82 × 10^4^ km^2^, respectively. Compared to the present, the total suitable area increased by 121.07 × 10^4^ km^2^, and the high suitability area increased by 73.20 × 10^4^ km^2^. The area of medium suitability first increased and then decreased, reaching its maximum during the LGM at 108.13 × 10^4^ km^2^, an increase of 50.15 × 10^4^ km^2^ compared to the present. The area of low suitability first decreased and then increased, reaching its maximum value during the LIG at 111.53 × 10^4^ km^2^, an increase of 13.02 × 10^4^ km^2^ compared to the low suitability area under the current climate scenario.

#### 3.3.2. Current Potential Distribution Estimates

Figure 5d and Table 2 and Table 3 indicate under current climate conditions, the total suitable area for *H. perforatum* is 188.11 × 10^4^ km^2^, and the high suitability areas are mainly distributed in Chengdu, Deyang, and Mianyang of Sichuan Province, Yubei District, Hechuan City, and Tongnan County of Chongqing, and the Ili region of Xinjiang, with a total distribution area of 31.62 × 10^4^ km^2^. The medium suitability areas are primarily located in Longnan, Tianshui, and Wudu of Gansu Province, Bijie, Liupanshui of Guizhou Province, and Tacheng of Xinjiang, with a total distribution area of 57.98 × 10^4^ km^2^. The low suitability areas are mainly distributed in Shaanxi Province, Henan Province, Hubei Province, Zhejiang Province, and Anhui Province, covering an area of 98.51 × 10^4^ km^2^.

#### 3.3.3. Suitable Distribution under Future Climate Scenarios

Figure 6 and Table 4 show under future climate scenarios, the total suitable area for *H. perforatum* shows an expanding trend. The high suitability areas are mainly in Chengdu, Meishan, Leshan, and Ya’an of Sichuan Province, and Tacheng and Ili of Xinjiang. The medium suitability are primarily located in Bijie, Liupanshui, and Qianxinan Buyi and Miao Autonomous Prefecture of Guizhou Province, and Longnan and Tianshui of Gansu Province. The low suitability areas are mainly found in Gansu, Ningxia, Shaanxi, Shanxi, Henan, Hebei, Sichuan, Guizhou, Hunan, Hubei, Anhui, Zhejiang, and Jiangxi.

Table 5 shows that under the 2050s-SSP245 climate scenario, the total suitable habitat area for *H. perforatum* reaches its maximum, expanding by 116.80 × 104 km^2^ compared to the present day. In the 2090s-SSP245 climate scenario, the total suitable habitat area ranks second at 223.38 × 10^4^ km^2^, an increase of 35.27 × 10^4^ km^2^ from the present. Compared to the present, under future climate scenarios, both the low and high suitability habitat areas are projected to increase, with the most significant increase in the high suitability habitat area occurring under the 2050s-SSP245 climate scenario, expanding by 68.93 × 10^4^ km^2^ and 5.24 × 10^4^ km^2^, respectively. In the 2090s-SSP585 climate scenario, the medium suitability habitat area is expected to undergo the greatest reduction, decreasing by 4.54 × 10^4^ km^2^ compared to the present.

Figure 6 and Table 4 reveal that under future climate scenarios, the suitable habitats for *H. perforatum* do not exhibit significant macroscopic changes. Under the SSP126, SSP370, and SSP585 climate scenarios, there are no notable fluctuations in the area covered by the four suitability categories for both the 2050s and 2090s periods. However, under the SSP245 climate scenario, there is a considerable difference in the total suitable habitat area between the 2050s and 2090s. The total suitable area in the 2050s is 304.91 × 10^4^ km^2^, an increase of 116.80 × 10^4^ km^2^ from the present day, while the total area in the 2090s is only 223.38 × 10^4^ km^2^, an increase of just 11.69 × 10^4^ km^2^ from the present. Moreover, the total suitable habitat area for *H. perforatum* under the SSP245 scenario is larger and more extensive compared to the other three climate scenarios.

### 3.4. Spatial Pattern Changes under Future Climate Conditions

In this study, ArcGIS 10.8 software was used to calculate the trend and range of changes in the suitable area for *H. perforatum* under different future climate scenarios compared to the contemporary area, obtaining the areas and geographical ranges of expansion, retention, and contraction, with the results presented in Table 6 and Figure 7.

The results show that, compared to the contemporary situation, the area of expansion under the 2050s-SSP585 climate scenario is the largest, followed by the 2090s-SSP126 climate scenario, with expansion areas of 31.06 × 10^4^ km^2^ and 29.78 × 10^4^ km^2^, respectively. The main expansion areas are concentrated in Suining, Zigong, and Neijiang of Sichuan Province; Lichuan, Yichang, and Enshi Tujia and Miao Autonomous Prefecture of Hubei Province; Tacheng and Altay of Xinjiang; Xiangxi Tujia and Miao Autonomous Prefecture, Zhangjiajie, and other regions. Under the 2090s-SSP585 and 2090s-SSP245 climate scenarios, the contraction areas are larger, with areas of 25.08 × 10^4^ km^2^ and 22.89 × 10^4^ km^2^, respectively. The contraction areas are mainly concentrated in Dazhou and Bazhong of Sichuan Province; Ningxia; Liuzhou of Guangxi; and in the Longnan and Tianshui regions of Gansu Province.

### 3.5. The Migration Trends of the Geometric Center of Suitable Habitat

To further explore the dynamic migration pathways of *H. perforatum*, this study used the SDM_Toolbox_v2.4 software to calculate the centroid latitude and longitude of the distribution of *H. perforatum* under current and different future climate scenarios. The results are shown in Figure 8 and Figure 9. This analysis concentrates the distribution of *H. perforatum* into a single central point and creates a vector file that describes the magnitude and direction of change over time. The trend of the distribution change in *H. perforatum* is analyzed by observing the changes in the centroid. Currently, the centroid of the suitable distribution area of *H. perforatum* is located in Yanxi Town, Bazhong City, Sichuan Province (107.239° E, 32.125° N).

Overall, from the past to the present, *H. perforatum* has migrated over a considerable distance; during the LIG period, the centroid of the suitable distribution area for *H. perforatum* was located in Zhenping County, Ankang City, Shaanxi Province (109.413° E, 31.876° N). During the LGM period, its centroid was situated in Pingchang County, Bazhong City, Sichuan Province (107.194° E, 31.713° N).

The latitude of the centroid of the suitable distribution area for *H. perforatum* during the LIG and LGM periods was lower than that of the contemporary suitable distribution area, while during the MH period, it was higher than the contemporary suitable distribution area, with different migration directions. From the LGM to the MH period, the migration distance was the farthest, at 292.398 Km. From the present to the future, with the exception of the 2090s-SSP126 and 2050s-SSP245 climate scenarios, where the distribution centroid of the suitable area for *H. perforatum* migrates to the southwest and to lower latitudes, other climate scenarios show a migration toward the northwest and to higher latitude regions.

## 4. Discussion

### 4.1. Rationality of Model

The MaxEnt model is a machine learning model based on the entropy theory proposed Jaynes in 1957 [35]. Subsequently, Phillips and others developed the MaxEnt 3.4.4 software based on the maximum entropy theory using the JAVA 5.0 programming in 2004. It only requires species presence (or occurrence) data and environmental information to run. Compared to other niche models, the MaxEnt model not only has good predictive performance and stability but also has excellent spatiotemporal transferability, simple and easily obtainable data types [36], simple and fast operation, and low sample requirements. It does not require excessive reliance on prior experience, has high prediction accuracy, and provides more intuitive result explanations. In addition, the MaxEnt model has a built-in Jackknife test to assess the significance of individual environmental variables, making it an ideal predictive tool for many researchers. In this study, we obtained information on 894 distribution points for *H. perforatum* and data on eight dominant ecological environmental factors. Based on MaxEnt model prediction results, it was inferred that the suitable regions for *H. perforatum* are mainly concentrated in the central and eastern region of China, such as Guangxi, Sichuan, Chongqing, Shandong, Gansu, Henan, Hunan, Hubei, and Guizhou provinces. These regions largely coincide with the current main distribution areas or primary production areas of *H. perforatum*, indicating that the predictions of the MaxEnt model are highly reliable.

### 4.2. Climate Effects

The environment is an important factor in the formation and variation in chemical substances in medicinal materials, and the quality and therapeutic efficacy of medicinal materials are closely related to their geographical distribution [37]. From the research results, it was found that the dominant environmental factors affecting the distribution of *H. perforatum* are April solar radiation (Srad4), September solar radiation (Srad9), the average temperature of the driest quarter (Bio9), November solar radiation (Srad11), annual mean temperature (Bio1), and annual precipitation (Bio12), which are consistent with the sun-loving characteristics of *H. perforatum* [38]. Temperature and precipitation have important impacts on species distribution, but due to different growth habits of species, the impact of each bioclimatic variable varies with the species. Kang studied the potential geographic distribution of *Lantana camara* in China, and the results showed that the dominant factors affecting the potential distribution pattern of *Lantana camara* are the Human Influence Index (HII), followed by the annual temperature range (Bio7) and solar radiation [39]. However, this experiment only considered climatic alpine and soil factors affecting the growth of *H. perforatum*, without considering factors such as human activities and variety selection. Sayti et al. found that human activities have a significant impact on the geographical distribution patterns of species [40]. After adding human activities as a variable, the MaxEnt simulation of the suitable area for *Solanum rostratum* decreased from 13.04% to 9.57%. However, the intensity of human activities is unclear in future changes, so it was not chosen. Therefore, it can be inferred that the suitable area for *H. perforatum* in this study’s results is larger than the actual suitable area.

### 4.3. Change in Geographic Distributions

The Intergovernmental Panel on Climate Change (IPCC) in its Sixth Assessment Report (AR6) released in 2021 proposed the Shared Socioeconomic Pathways (SSPs) which indicate that climate change, temperature rise, and increased precipitation are becoming more pronounced. Thomas et al. found that under moderate CO_2_ emission concentrations in the 2050s, 15–37% of species could face the risk of extinction, while other species would face a lower risk of extinction, and some species might benefit from global warming, suggesting that the impact of climate warming on the potential geographical distribution of species is two-sided [41]. The results of this study show that under future climate scenarios, the total suitable area for *H. perforatum* is on an expanding trend. The northeastern Yunnan, western Hubei, and central Sichuan in China will become more suitable for the growth of *H. perforatum*, while central Guizhou, western Henan, and other regions will no longer be suitable for its growth. Leng et al. found that under future climate change scenarios, the potential geographic distribution patterns of three species of deciduous larches would significantly shift toward higher latitude regions [42]. Chen et al. found through meta-analyses of species distribution studies that species distributions may shift towards higher latitudes at an average rate of 16.9 km per decade [43]. In this study, overall, under future climate scenarios, the distribution center of *H. perforatum* is shifting towards higher latitude regions, which is consistent with previous research.

## 5. Conclusions

The application of the MaxEnt model and ArcGIS technology can effectively simulate the suitable distribution areas for *H. perforatum* in China and identify the dominant climatic factors affecting its distribution as April solar radiation (Srad4), September solar radiation (Srad9), average temperature during the driest quarter (Bio9), November solar radiation (Srad11), annual mean temperature (Bio1), and annual precipitation (Bio12). Comparing the past and the future, the highly suitable areas for *H. perforatum* are relatively stable and are mainly distributed in Chengdu, Meishan, Leshan, and Ya’an in Sichuan and Tacheng, Ili in Xinjiang; the moderately suitable areas are mainly in Bijie, Liupanshui, and Qianxinan Buyi and Miao Autonomous Prefecture in Guizhou, Longnan and Tianshui in Gansu. In the future, the suitable area for *H. perforatum* is expected to expand to some extent, with new areas mainly concentrated in northeastern Yunnan, western Hubei, central Sichuan, Tacheng and Altai in Xinjiang, while areas experiencing a reduction in suitability are mainly in central Guizhou, western Henan, and other regions. The centroid of its distribution is expected to shift towards the northwest and higher latitude regions under future climate scenarios.

## Figures and Tables

**Figure 1 biology-13-00452-f001:**
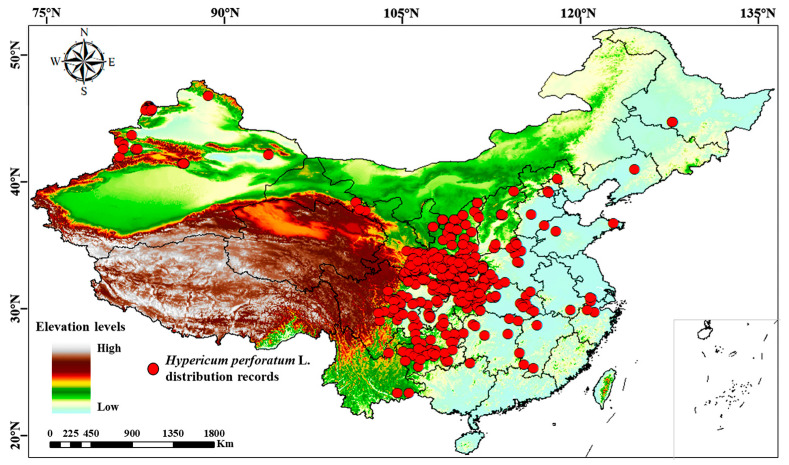
Geographical distribution of *H. perforatum* in China. Note: Red points represent species distribution records.

**Figure 2 biology-13-00452-f002:**
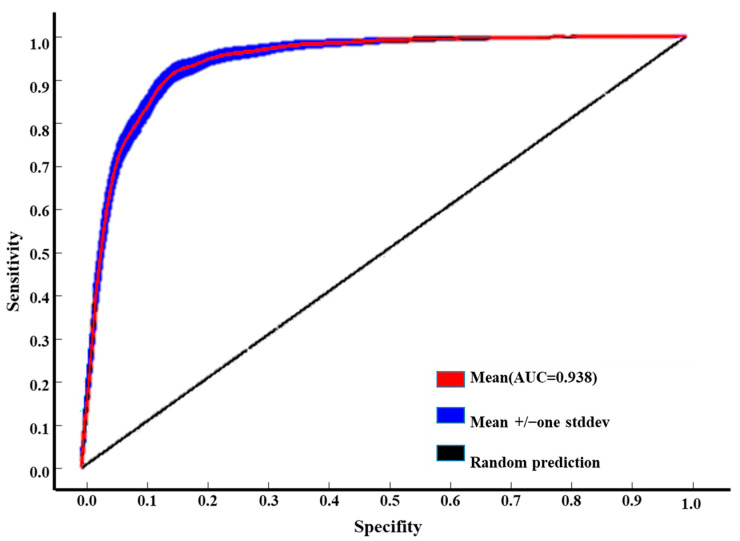
AUC of current of *H. perforatum* using ROC methods to test the results of MaxEnt.

**Figure 3 biology-13-00452-f003:**
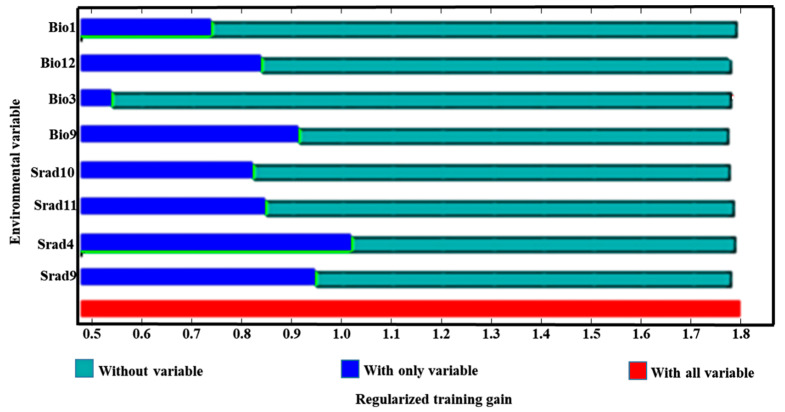
Results of Jackknife tests for the contribution of variables to the habitat distribution model of *H. perforatum* for at present. The regularized training gain describes how well the MaxEnt distribution fits the presence data compared to a uniform distribution. The dark blue bars indicate the gain from using each variable in isolation, the light blue bars indicate the gain lost by removing the single variable from the full model, and the red bar indicates the gain using all of the variables.

**Figure 4 biology-13-00452-f004:**
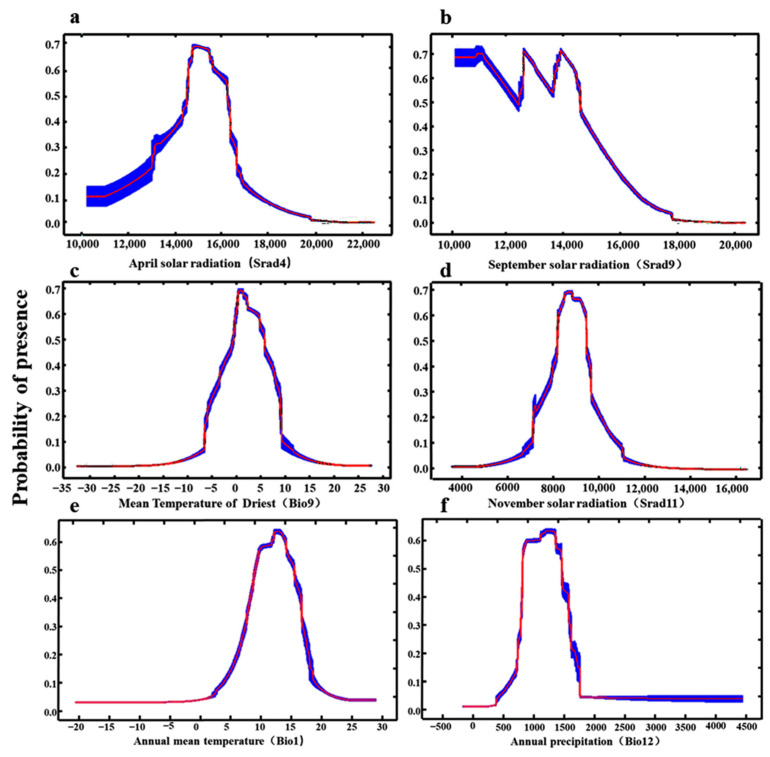
Singlefactor response curve of the current existence probability to the bioclimatic variables. (**a**) April solar radiation; (**b**) September solar radiation; (**c**) Mean Temperature of Driest; (**d**) November solar radiation; (**e**) Annual mean temperature; (**f**) Annual precipitation.

**Figure 5 biology-13-00452-f005:**
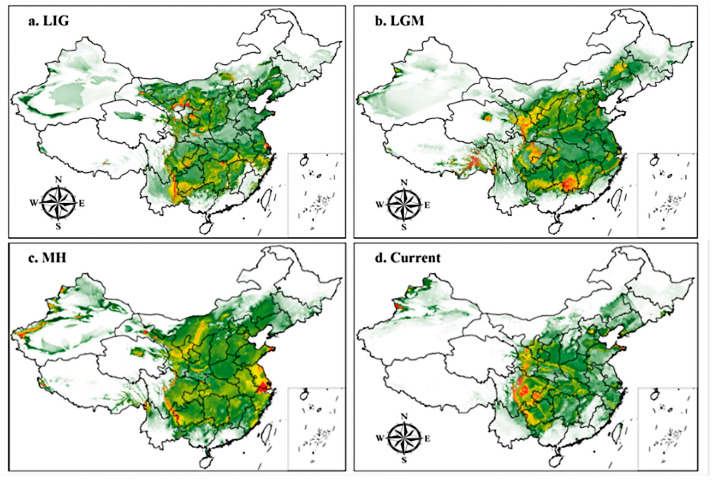
Potential distribution for *H. perforatum* in China under past and current climate conditions predicted by the MaxEnt model. The potential distribution of *H. perforatum* was divided into four grades by the natural breaks method. Red, yellow, green, and white areas represent highly suitable, moderately suitable, marginally suitable, and not suitable areas, respectively. (**a**) LIG; (**b**) LGM; (**c**) MH; (**d**) Current.

**Figure 6 biology-13-00452-f006:**
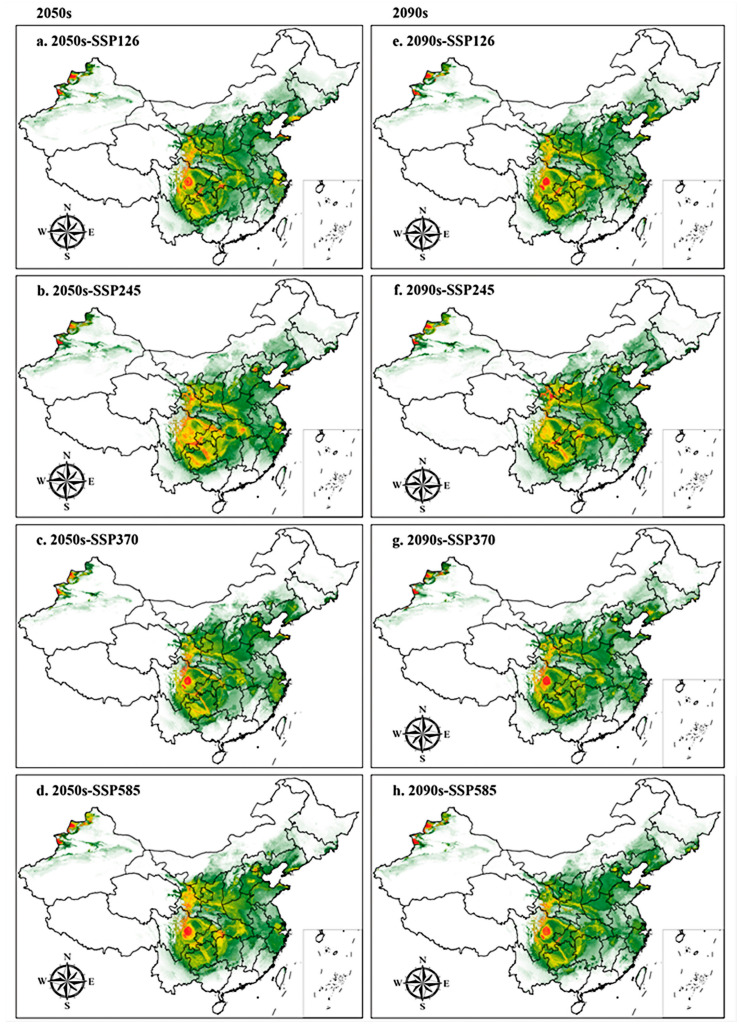
Potential distribution for *H. perforatum* in China under future climate conditions predicted by the MaxEnt model. The potential distribution of *H. perforatum* was divided into four grades by the natural breaks method. Red, yellow, green, and white areas represent highly suitable, moderately suitable, marginally suitable, and not suitable areas, respectively.(**a**) 2050s-SSP126; (**b**) 2050s-SSP245; (**c**) 2050s-SSP370; (**d**) 2050s-SSP585; (**e**) 2090s-SSP126; (**f**) 2090s-SSP245; (**g**) 2090s-SSP370; (**h**) 2090s-SSP585.

**Figure 7 biology-13-00452-f007:**
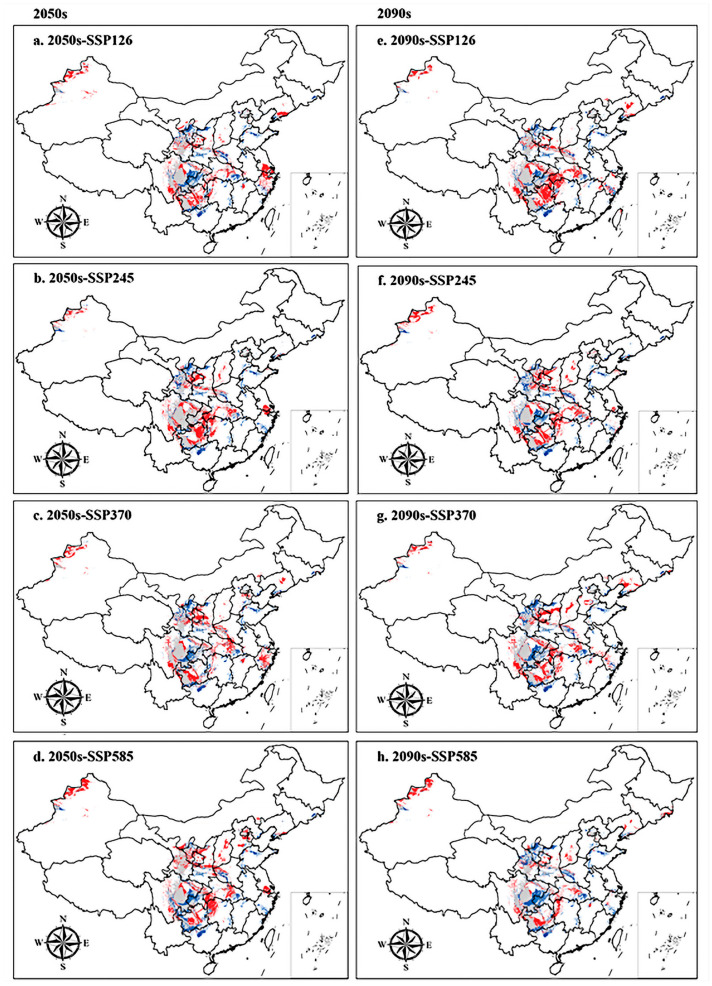
Spatial changes in *H. perforatum* in China under emission scenarios of the 2050s and 2090s. White, grey, red, and blue areas represent not suitable, unchanged suitable, expansion suitable, and contraction suitable areas, respectively. (**a**–**d**), the 2050s; (**e**–**h**), the 2090s; (**a**,**e**), future climate scenario SSP126; (**b**,**f**), future climate scenario SSP245; (**c**,**g**), future climate scenario SSP370; (**d**,**h**), future climate scenario SSP585.

**Figure 8 biology-13-00452-f008:**
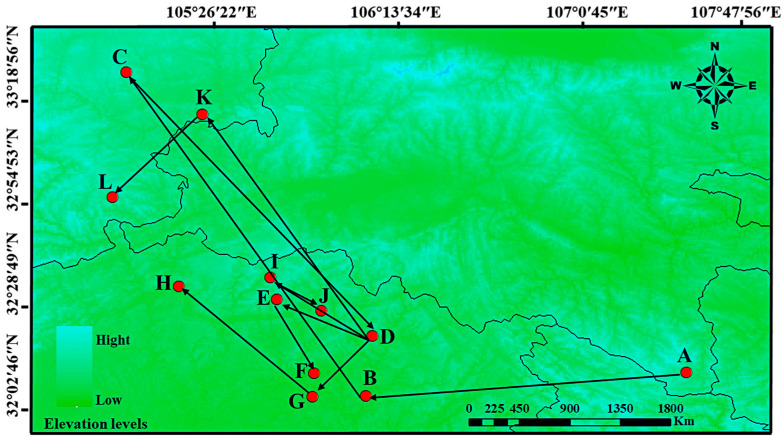
Migration of the center of suitable habitat for *H. perforatum* migratory routes in historical and future climate scenarios. Arrows indicate migratory routes and direction of the suitable habitat distribution center under historical and future climate scenarios. Among them, the meaning of the letters are as follows: (A) LIG; (B) LGM; (C) MH; (D) current; (E) 2050s-SSP126; (F) 2090s-SSP126; (G) 2050s-SSP245; (H) 2090s-SSP245; (I) 2050s-SSP370; (J) 2090s-SSP370; (K) 2050s-SSP585; (L) 2090s-SSP585.

**Figure 9 biology-13-00452-f009:**
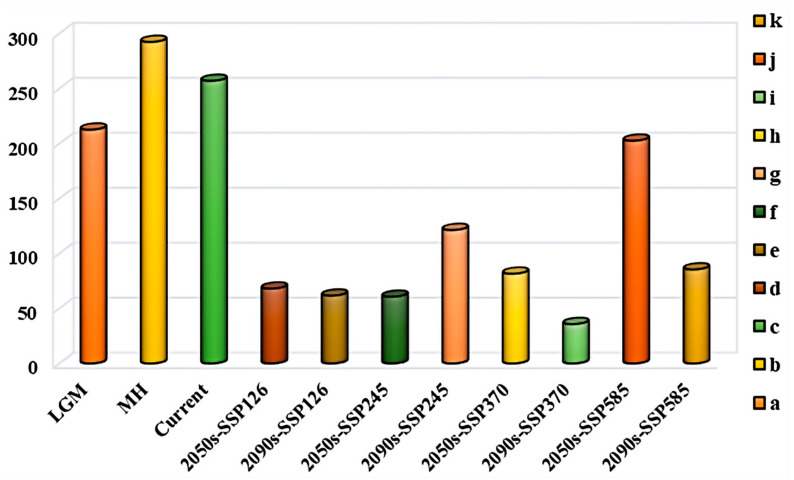
Centroid migration distance for *H. perforatum* under different scenarios/years. Among them, the meaning of the letters are as follows: (a) LIG→LGM; (b) LGM→MH; (c) MH→ Current; (d) Current→2050s-SSP126; (e) 2050s-SSP126→2090s-SSP126; (f) Current→2050s-SSP245; (g) MH; (h) Current→2050s-SSP370; (i) 2050s-SSP370→2090s-SSP370; (j) Current→2050s-SSP585; (k) 2050s-SSP585→2090s-SSP585.

**Table 1 biology-13-00452-t001:** The percentage contributions and permutation importance for environmental variables in the Maxent model.

Num.	Variable	Percent Contribution (%)	Permutation Importance (%)
1	April solar radiation (Srad4)	22.50	1.80
2	September solar radiation (Srad9)	21.60	13.70
3	Mean temperature of driest quarter (Bio9)	18.50	43.70
4	Solar radiation in November (Srad11)	13.20	10.40
5	October solar radiation (Srad10)	9.70	5.60
6	Annual mean temperature (Bio1)	7.00	9.80
7	Isothermality (Bio3)	5.60	4.70
8	Annual precipitation (Bio12)	2.00	10.20

**Table 2 biology-13-00452-t002:** Regional information of different suitable levels of *H. perforatum* in different periods.

Period	Regional Information		
	High Suitable Areas	Medium Suitable Areas	Low Suitable Areas
LIG	Yinchuan City, Ningxia Hui Autonomous Prefecture	Ji’an City, Jiangxi Province	Shaanxi, Shanxi, Liaoning, Henan, Hubei, Anhui
LGM	Yongzhou City, Hengyang City, Hunan Province	Wuzhong City, Ningxia Hui Autonomous Prefecture	Shanxi, Hebei, Henan, Hubei, Anhui, Jiangsu
MH	Suzhou City, Jiangsu Province, Huzhou City, Jiaxing City, Zhejiang Province	Xuzhou City, Suqian City, Huai’an City, Yangzhou City, Jiangsu Province	Shanxi, Hebei, Henan, Hubei, Anhui
Current	Chengdu City, Deyang City, Mianyang City, Sichuan Province Yubei District, Hechuan City, Tongnan City, Chongqing City	Longnan City, Tianshui City, Wudu City, Luizhou Province; Bijie City, Liupanshui City, Guizhou Province; Tacheng District, Xinjiang Uygur Autonomous Region	Shaanxi, Henan, Hubei, Zhejiang, Anhui

**Table 3 biology-13-00452-t003:** Characteristics of potential distribution in different periods for *H. perfotum*.

Period	Area of Each Suitable Habitat (×10^4^ km^2^)				
	Non Suitable Areas	Low Suitable Areas	Medium Suitable Areas	High Suitable Areas	Total Suitable Areas
LIG	650.82 (−121.07)	111.53 (+13.02)	92.84 (+34.86)	104.82 (+73.20)	309.18 (+121.07)
LGM	714.91 (−56.89)	67.67 (−30.84)	108.13 (+50.15)	69.3 (+37.68)	245.09 (+56.98)
MH	766.05 (−5.84)	97.83 (−0.68)	62.74 (+4.76)	33.38 (+1.76)	193.95 (+5.84)
Current	771.89 (0.00)	98.51 (0.00)	57.98 (0.00)	31.62 (0.00)	188.11 (0.00)

Note: The numbers in parentheses represent the change in the area of suitable habitats for *H. perforatum* compared to the current, with “+” indicating an increase in area and “−” indicating a decrease.

**Table 4 biology-13-00452-t004:** Regional information of different suitable levels of *H. perfotum* in different scenarios.

Climate Scenarios	Regional Information		
	High Suitable Areas	Medium Suitable Areas	Low Suitable Areas
SSP126	Chengdu City, Meishan City, Leshan City, Ya’an City, Sichuan Province; Tacheng District, Yili District, Xinjiang	Liupanshui City, Guizhou Province	Shaanxi, Shanxi, Henan, Hunan
SSP245	Pingliang City, Tianshui City, Gansu Province	Chengdu City, Ya’an City, Meishan City, Leshan City, Sichuan Province	Hebei, Henan, Shandong, Zhejiang
SSP370	Chengdu City, Meishan City, Leshan City, Ya’an City, Sichuan Province; Tacheng District, Yili District, Xinjiang	Liupanshui City, Guizhou Province	Hebei, Henan, Shandong, Zhejiang
SSP585	Chengdu City, Meishan City, Leshan City, Ya’an City, Sichuan Province; Tacheng District, Yili District, Xinjiang	Tianshui City, Longnan City, Gansu Province	Henan, Shandong, Anhui, Zhejiang

**Table 5 biology-13-00452-t005:** Characteristics of potential distribution in future periods for *H. perforatum*.

Period	Area of Each Suitable Habitat (×10^4^ km^2^)				
	Non Suitable Areas	Low Suitable Areas	Medium Suitable Areas	High Suitable Areas	Total Suitable Areas
Current	771.89 (0.00)	98.51 (0.00)	57.98 (0.00)	31.62 (0.00)	188.11 (0.00)
2050s-SSP126	755.81 (−16.08)	115.34 (+16.83)	57.01 (−0.97)	31.84 (+0.22)	204.19 (+16.08)
2050s-SSP245	655.08 (−116.81)	167.44 (+68.93)	100.61 (+42.63)	36.86 (+5.24)	304.91 (+116.80)
2050s-SSP370	758.56 (−13.33)	114.4 (+15.89)	55.24 (−2.74)	31.80 (+0.18)	201.44 (+13.33)
2050s-SSP585	759.67 (−12.22)	111.53 (+13.02)	56.97 (−1.01)	31.83 (+0.21)	200.33 (+12.22)
2090s-SSP126	763.66 (−8.23)	108.25 (+9.74)	56.45 (−1.53)	31.65 (+0.03)	196.34 (+8.23)
2090s-SSP245	628.83 (−143.06)	174.28 (+58.40)	61.84 (+6.11)	33.61 (+1.99)	223.38 (+35.27)
2090s-SSP370	754.29 (−17.60)	117.37 (+18.86)	53.88 (−4.1)	34.46 (+2.84)	205.71 (+17.6)
2090s-SSP585	757.94 (−13.95)	116.69 (+18.8)	53.44 (−4.54)	31.94 (+0.32)	202.06 (+13.95)

**Table 6 biology-13-00452-t006:** Spatial changes in the suitable growing areas of *H. perforatum* under future climate scenarios.

Period	Area (×10^4^ km^2^)			
	Expansion Zone	Reserve Zone	Shrinkage Zone	Changes
2050s-SSP126	25.91	33.25	22.33	3.58
2050s-SSP245	27.33	37.76	17.81	9.52
2050s-SSP370	26.38	35.04	20.37	6.01
2050s-SSP585	31.06	33.15	22.42	8.64
2090s-SSP126	29.78	36.26	19.31	10.47
2090s-SSP245	26.44	33.01	22.57	3.87
2090s-SSP370	26.59	32.68	22.89	3.70
2090s-SSP585	18.48	30.50	25.08	−6.60

## Data Availability

The data presented in this study are available from the authors upon request.

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
