# Peer review of "Predicting the Potential Distribution of Hypericum perforatum under Climate Change Scenarios Using a Maximum Entropy Model"

_biology, 2024, doi:10.3390/biology13060452_

Round 1

Reviewer 1 Report

Comments and Suggestions for Authors

Overall, the article provides valuable insights into the geographic areas and climatic factors influencing the distribution of Hypericum perforatum. However, there is a need for improvement in the figures and table legends. 

The figures and table legends should be revised to provide more clarity and detail. Currently, they lack sufficient information to fully understand the data being presented. It would be helpful to include more specific explanations and labels to guide readers in interpreting the figures and tables accurately.

Additionally, the title of the article should be revised to better reflect the content and focus of the study. The current title is too general and does not adequately convey the specific topic being discussed.

Furthermore, the article should delve into the medicinal properties and health benefits of Hypericum perforatum. This information is mentioned briefly in the introduction but is not explored in depth throughout the rest of the article. Including more detailed explanations of the medicinal properties and health benefits would enhance the overall value of the article.

Moreover, the article should provide more information on the various species and subspecies of Hypericum perforatum. Currently, the focus is primarily on the geographic distribution and climatic factors, but there is limited discussion on the different types of Hypericum perforatum. Including more information on the various species and subspecies would provide a more comprehensive understanding of the plant.

The figures depicting the geographic distribution of Hypericum perforatum require further elaboration. Currently, they are not adequately explained, and it is difficult to discern the specific regions being represented. Providing more detailed explanations and labels for the geographic distribution figures would improve their clarity and usefulness.

Similarly, the charts and graphs in the article need more detailed explanations. Currently, they are not fully explained, and it is unclear what specific data they are presenting. Including more detailed explanations and labels for the charts and graphs would help readers understand the data being presented.

Lastly, some regions are vaguely mentioned throughout the article, and it would be beneficial to organize this information in a tabular or list format. This would make it easier for readers to reference and understand the different regions discussed in the article.

Overall, the article provides valuable insights into the geographic distribution and climatic factors influencing Hypericum perforatum. However, improvements are needed in the figures and table legends, as well as more detailed explanations to help readers understand the finer points. Additionally, delving into the medicinal properties and health benefits of the plant, providing more information on the various species and subspecies, and further elaborating on the geographic distribution and charts and graphs would enhance the overall quality and value of the article.

Comments on the Quality of English Language

The article on the quality of the English language is generally well-written with minimal grammatical mistakes. The author demonstrates a strong command of the language, employing a wide range of vocabulary and sentence structures to effectively convey their ideas. The article flows smoothly, with coherent paragraphs and logical transitions between different points.

Author Response

Thank you for your valuable comments. In this revision, I have made the following changes to the article:

(1) Provided more detailed explanations and descriptions for the figures and tables in the article;

(2) Modified the title of the article;

(3) Organized and analyzed the regions mentioned in the article in the form of tables;

(4) Supplemented the species information and pharmacological effects of Hypericum perforatum.

Once again, thank you for your valuable comments. I hope my explanations can address your concerns. The specific modifications will be uploaded in the form of a document.

Reviewer 2 Report

Comments and Suggestions for Authors

This paper uses MaxEnt to predict the potential suitable climates for Hypericum perforatum under climate change. The modeling approach is standard, and the paper is mostly well written. I have a few comments that I would like the authors to address. 

First, justification for using predictor variables. The authors have not given any specific reasons for choosing the predictors that they have used in the modelling. They make somewhat generic statements in the Discussion, but I don’t see anything concrete that ties the distribution of this plant to climate, topography, and soil characteristics. The relationships then would purely be data-driven and not based on biological relevance. 

Second, I am wondering why the authors used climate data from historical time periods (thousands to 120,000 – 140,000 years ago!). I don’t see any reason or justification for using them. Just simply, the current and future time periods would have been enough. 

Finally, did the authors try some other measure to test the performance of the model such as TSS (true skill statistic)? AUC is not considered a reliable measure (Lobo et al. 2007 Global Ecology and Biogeography). 

Minor comments

Avoid using “hot topic” or “hot research topic”.

Line 166: Generally, more than one climate model is used in species distribution modelling to account for variability in future climate projections. Did the authors consider other climate models in addition to the one used in the study?

Line 274: ‘Suitable’

It would help readers if a map showing some of the location names used in the manuscript (e.g. 277-279) is included. Most will not be familiar with the locations mentioned in the manuscript.

Figure 9 is confusing. What is the measure on the y-axis? ‘5026’, ‘9026’, etc., are very confusing. Please modify.

Check references as some of them are not in sentence case and latin names are not italicized (lines 527, 540, 552, 566).

Comments on the Quality of English Language

No comments.

Author Response

Thank you for your valuable comments. In this revision, I have made the following changes to the article:

(1)The reasons for selecting climate, terrain, and soil as predictive variables in this study are as follows: Medicinal plants, as a crucial source of traditional Chinese medicine, play a significant role in the fight against diseases. The growth, development, reproduction, and ecological distribution of each medicinal plant are closely related to ecological factors such as climate, soil, and terrain. Suitable ecological factors are conducive to the growth and development of medicinal plants, as well as the production and accumulation of active ingredients, thereby improving the quality of medicinal materials. Among these factors, temperature serves as the heat source for medicinal plants during seed germination and subsequent growth and development. Water is essential for maintaining normal photosynthesis and other metabolic processes in plants. Insufficient changes in light intensity and photoperiod can lead to poor plant development. Soil is also an indispensable condition for the growth of medicinal plants. The chemical properties of soil are major factors affecting the availability of soil nutrients and the presence of chemical substances in the soil.

(2) This study uses the AUC value as the standard for evaluating the accuracy of the MaxEnt model, primarily referencing the following literature: â‘ Dong PB, Wang LY, Wang LJ, Jia Y, Li ZH, Bai G, Zhao RM, Liang W, Wang HY, Guo FX, Chen Y. Distributional Response of the Rare and Endangered Tree Species Abies chensiensis to Climate Change in East Asia. Biology (Basel). 2022 Nov 13;11(11):1659. â‘¡Cao YT, Lu ZP, Gao XY, Liu ML, Sa W, Liang J, Wang L, Yin W, Shang QH, Li ZH. Maximum Entropy Modeling the Distribution Area of Morchella Dill. ex Pers. Species in China under Changing Climate. Biology (Basel). 2022 Jul 8;11(7):1027.â‘¢Xiao F, Liu Q, Qin Y. Predicting the Potential Distribution of Haloxylon ammodendron under Climate Change Scenarios Using Machine Learning of a Maximum Entropy Model. Biology (Basel). 2023 Dec 20;13(1):0.

(3) For predicting the past suitable habitats of H. perforatum, the Community Climate System Model (CCSM) was used. This model is one of the new generation coupled climate system models internationally. CCSM4.0 was released in 2010, and its overall performance has seen significant improvements compared to previous versions .CCSM4.0 consists of five modules: atmosphere, ocean, land surface, sea ice, and coupler. The model is open-source, supports multiple resolutions, and allows for different combinations of modules to meet the needs of this study. (Gent P R, Danabasoglu G, Donner L J, et al. 2011. The community climate system model version. J. Climate, 24 (19): 4973–4991.)。When predicting the future suitable habitat for H. perforatum, the BCC-CCSM2-MR climate model is used. The global climate model BCC-CSM2-MR (Beijing Climate Center-Climate System Model version 2-Medium Resolution) was independently developed by the National (Beijing) Climate Center and participated in the sixth phase of the Coupled Model Intercomparison Project. BCC-CSM2-MR has improved the simulation capability of seasonal average precipitation in most regions of East Asia, especially the summer average precipitation in the Tibetan Plateau. It has significantly enhanced the simulation performance of monthly precipitation variations in southeastern China, the Korean Peninsula, and Japan, and has markedly improved the simulation capability of daily extreme precipitation (temperature) in southeastern China. (Li Shuping, Quan Wenjie, Wang Zheng, et al. Evaluation of BCC-CSM2-MR Global Climate Model on Precipitation and Temperature Simulation in East Asia. Arid Meteorology, 2023, 41(06): 984-996.)

(4) There are several important reasons for using historical climate data (from thousands to 120,000–140,000 years ago) in this study: â‘  Understanding ecological adaptability: Analyzing historical climate data helps to reveal the survival and distribution patterns of H. perforatum in the context of long-term climate change. â‘¡ Predicting climate change patterns: Utilizing past climate change patterns can improve the accuracy of our predictions for future changes. â‘¢ Biodiversity conservation: By identifying the impact of past climate changes on the distribution of Hypericum perforatum, we can develop more effective conservation measures to address future climate changes.

Thank you again for your valuable feedback. I hope my explanation addresses your concerns. I will upload the specific modification information in the form of a document.
